# Unsustainable anthropogenic mortality threatens the long-term viability of lion populations in Mozambique

João Almeida[1‡], Willem D. Briers-Louw[2,3‡*], Agostinho Jorge[4], Colleen Begg[4], Marnus Roodbol[5], Hans Bauer[6], Andrew Loveridge[6,7], Matthew Wijers[6], Rob Slotow[8], Peter Lindsey[9,10], Kristoffer Everatt[7,11], Holly Rosier[12], Sean Nazerali[13], Lizanne Roxburgh[5], Hugo Pereira[1], Mercia da Conceicao[14], Armindo Araman[15], Osvaldo J. Abrao[5,16], Alison J. Leslie[2], Franziska Steinbruch[17], Vincent N. Naude[2], Samantha K. Nicholson[5]

1 Mozambique Wildlife Alliance, Maputo, Mozambique, 2 Department of Conservation Ecology and Entomology, University of Stellenbosch, Stellenbosch, South Africa, 3 Zambeze Delta Conservation, Marromeu, Sofala, Mozambique, 4 Niassa Carnivore Programme, Mariri/ TRT Conservation Foundation, Niassa Special Reserve, Mozambique, 5 The Endangered Wildlife Trust, Conservation Science Unit, Johannesburg, South Africa, 6 Wildlife Conservation Research Unit, Recanati-Kaplan Centre, Department of Biology, University of Oxford, Tubney, United Kingdom, 7 Panthera, New York City, New York, United States of America, 8 Centre for Functional Biodiversity, School of Life Sciences, University of KwaZulu-Natal, Durban, South Africa, 9 Mammal Research Institute, Department of Zoology, University of Pretoria, Pretoria, South Africa, 10 Wildlife Conservation Network, San Francisco, California, United States of America, 11 Greater Limpopo Carnivore Programme, Limpopo National Park, Gaza, Mozambique, 12 Rio Save Safaris, Manica, Mozambique, 13 Independent researcher, Maputo, Mozambique, 14 Gorongosa National Park, Sofala, Mozambique, 15 Administração Nacional das Áreas de Conservação, Maputo, Mozambique, 16 Peace Parks Foundation, Stellenbosch, South Africa 17 Planether, Maputo, Mozambique

‡ JA and WBL shared first co-authorship on this work.
* 17193729@sun.ac.za

## Abstract

Anthropogenic mortality is a pervasive threat to global biodiversity. African lions (*Panthera leo*) are particularly vulnerable to these threats due to their wide-ranging behaviour and substantial energetic requirements, which typically conflict with human activities, often resulting in population declines and even extirpations. Mozambique supports the 7th largest lion population in Africa, which is recovering from decades of warfare, while ongoing conflicts and broad-scale socio-economic fragility continue to threaten these populations. Moreover, there are concerns that Mozambique represents a regional hotspot for targeted poaching of lions which fuels a transnational illegal wildlife trade. This study aimed to quantify the longitudinal impact of anthropogenic mortality on lion populations in Mozambique. Using national population estimates and monitoring records, we performed forward simulation population viability modelling incorporating detection-dependent population trends and varying scales of anthropogenic mortality. Between 2010–2023, 326 incidents of anthropogenic mortality involving 426 lions were recorded. Bushmeat bycatch and targeted poaching for body parts were the greatest proximate causes of lion mortality (i.e., 53% of incidents),

**Data availability statement:** All relevant data are within the manuscript, its Supporting Information file and data repository (https://doi.org/10.6084/m9.figshare.29040002).

**Funding:** The author(s) received no specific funding for this work.

increasing significantly over time and acting as cryptic suppressors of regional population recovery, followed by legal trophy hunting (i.e., 33%), and retaliatory killing (i.e., 13%). Our findings suggest that resilience to anthropogenic threats is largely a function of lion population size as well as resource and management capacity. For instance, projections suggest that the lion population in Niassa Special Reserve will likely remain stable despite comparatively high levels of anthropogenic mortality, although further escalation may precipitate decline. Conversely, the lion population in Limpopo National Park is projected to become extirpated by 2030 without the buffering effect of its neighbouring source population in Kruger National Park. These unsustainable levels of anthropogenic mortality threaten the long-term viability of lion populations in Mozambique, requiring urgent national-level action and public-private partnerships to support site security, monitoring, and policy enforcement.

## Introduction

Escalating anthropogenic mortality arguably poses the greatest threat to global biodiversity [1], portending a sixth mass extinction crisis [2], with significant impacts on vital ecological and socio-economic services [3]. In Africa, anthropogenic threats to wildlife are severe and pervasive at the continental (e.g., land/sea-use change, natural resource extraction, pollution, climate change [4,5]), regional (e.g., illegal wildlife trade [IWT], poorly regulated legal harvest [6]), and local (e.g., poaching [7] and human-wildlife conflict [HWC; 8]) scales. Understanding the spatio-temporal patterns of such direct and indirect mortality, and how current or planned policy may potentially mitigate these scale-dependent threats, is critical for effective monitoring and intervention strategies [9,10]. This applies particularly to large carnivores, which are inherently elusive and often occur at low densities, while their substantial metabolic and ranging requirements tend to exacerbate persecution and conflict [11,12], which often results in significant demographic impacts, population declines and even extirpations [7,13,14].

African lions (*Panthera leo*) are ecologically, economically, and culturally significant [12,15,16], and yet, these large carnivores continue to suffer widespread population and range declines [17–20], with an estimated 23,000 mature individuals occurring in ~6% of their historic range [17,21]. Despite being listed as 'Vulnerable' globally (International Union for Conservation of Nature Red List of Threatened Species [17]), successful lion conservation resulted in a ~12% increase between 1993 and 2014 across Botswana, Namibia, South Africa and Zimbabwe, largely due to the proliferation of reintroductions into relatively small (i.e., < 1,000 km²), fenced, and well-funded metapopulation reserves [18,22]. Nevertheless, this is not the case for most lion range states, where protected areas (PAs) are relatively large (i.e., > 1,000 km²), unfenced, and underfunded [23]. These range states are often unable to keep pace with mounting anthropogenic threat levels, experiencing a ~60% decline over the same period, with a notable increase in anthropogenic mortalities [18]. Unfortunately, similar declines are predicted for many lion populations across their African range,

especially in the most ecologically and socio-politically fragile countries [23,24], which may be without adequate resources and management capacity to ameliorate these threats [18,25].

Mozambique (~800,00km$^2$) is an important range state for lions in Africa, harbouring ~1,500 mature individuals [African Lion Database, unpublished data, 2024] and represents one of only nine African countries with >1,000 lions (i.e., 7[th] largest African lion population). With 29% of the country formally designated for conservation [26], there is an opportunity for population recovery and range restoration. However, in the shadow of historical armed conflict and political instability, Mozambican conservation continues to grapple with ongoing localized conflict, human settlement within or around PAs and unprecedented growth of resident communities. When coupled with severe and ongoing shortages in funding for PA management, these challenges have driven unsustainable levels of resource extraction [27,28]. Exacerbated by a rapidly growing human populus, post-war large carnivore and prey numbers have continued to decline across many parts of Mozambique [29–32]. Lions currently occur in six resident populations across Mozambique. Of these populations, Niassa Special Reserve (SR) in the northern region is considered the only lion stronghold (i.e., >500 lions; [17,33]) in the country, with stable to increasing populations occurring in the central region, largely due to post-war population recovery or subsequent reintroductions and significant private or non-governmental investment [30,31]. Mozambican lion populations are largely restricted to PAs and hunting concessions [34], and while these landscapes are relatively large, most are not sufficiently protected, monitored, or managed, as a result of limited resources and management capacity [20,23]. Mozambique also appears to have become a regional hotspot for IWT in lion derivatives, both as a source and transit state [35,36], that is particularly vulnerable to corruption and poor enforcement [14,37]. While the precise drivers of demand remain speculative, studies suggest that lion body parts are typically sought for use in zootherapeutic traditional medicine (i.e., zootherapy), and social or cultural purposes (e.g., clothing, curios, ornaments, trinkets, and other regalia) in both Asian and African markets [35,36,38]. Poaching of lions for body parts has thus raised significant concerns for lion conservation throughout the region [38,39].

Determining population size, as well as the proximate cause and extent of anthropogenic mortality for lions across Mozambique presents complex methodological and logistical challenges [36,40]. Approaches to estimating lion population size vary from non-individual-based techniques including traditional (e.g., call ups and spoor counts [41]) and more recent density estimation approaches (e.g., random encounter models [42]) to individual-based capture-recapture frameworks (e.g., search encounter [43]). Traditional approaches remain the most commonly applied across lion range states and provide sufficient information for estimating population trends [44]. Although, such methods carry inherent detection limitations and biases in addition to the substantial skilled capacity and operational costs of intensively monitoring such vast landscapes [43,45]. Similarly, the comparable detection of anthropogenic mortalities requires direct and consistent monitoring efforts [14,29,36,38], such that rapid and evidence-based reporting can distinguish among the proximate causes of lion mortality (i.e., natural, bushmeat bycatch, retaliatory killing, targeted poaching to supply the IWT, or legal trophy hunting). While regional consolidation and interpretation of such trends is objectively valuable, limited resources and capacity often force PAs and statutory authorities to prioritize immediate challenges like HWC and law enforcement over addressing knowledge gaps for adaptive management. Indirectly, PA management is also responsible for ensuring local food security and socio-economic development, where such resilience affords a greater capacity to coexist with lions and incentivizes alternative economic opportunities outside of poaching [23].

Understanding how lion populations are affected by anthropogenic mortality is critical to addressing this current conservation crisis and developing regional-level intervention strategies. This study aims to quantify the threat of anthropogenic mortality to the long-term viability of lion populations in Mozambique. Using longitudinal population estimates and mortality records collected between 2010–2023, we describe the recent trends and impact of anthropogenic mortality on current and future national lion population projections. We hypothesise that the proportion of anthropogenically linked lion mortality in Mozambique directly threatens the long-term viability of resident populations. We also predict that relative protected area size and perceived threat, as well as management resources and capacity contribute to the resilience of lion these

populations. While the targeted poaching of lions for the illegal wildlife trade significantly contributes to resident lion extirpation risk. We discuss our findings in the context of anthropogenic mortality relative to PA size, lion population viability, the detectability of mortalities and the scale of lion derivative trade, while evaluating resource and management capacity towards mitigating these threats across Mozambique. Importantly, we identify prioritized lion populations for strategic conservation management intervention and regional policy development.

## Methods

### Study areas

This study reports on longitudinal lion population and mortality data collected from various areas prioritized for conservation in Mozambique (Table 1). Of the national terrestrial conservation estate, formal PAs (i.e., national parks [NPs],

**Table 1. Summary of lion populations in Mozambique and anthropogenic mortality record.** Indicated by population, is the relevant management type (i.e., PA = Protected Area, WMA = Wildlife Management Area, MUL = Mixed-Use Landscape), area size, whether there is active trophy hunting, the latest population estimate and trend, the reported annual anthropogenic mortality and trend, as well as predominant causes of mortality and important contextual notes.

| Population | Management type | Area size (km²) | Trophy hunting of lions | Latest resident population estimate | Lion population trend | Anthropogenic mortality annual trend | Average annual anthropogenic mortalities | Main causes of anthropogenic lion mortality | Notes |
|---|---|---|---|---|---|---|---|---|---|
| **Banhine National Park** | PA | 7,290 | Inactive | Unknown | Unknown | No data | 2 | Illegal wildlife trade | *NA* |
| **Coutadas 9 & 13** | WMA | 9,724 | Active | 44 [1] | Increasing | Increasing | 2.72 | Poaching | Coutada 13 is considered a transient population |
| **Gorongosa National Park** | PA | 3,708 | Inactive | 183 (163-200) [2] | Increasing | Constant | 3.13 | By catch in snares | *NA* |
| **Greater Lebombo Conservancy** | WMA | 2,483 | Active | 70 [3] | Stable | Constant | No data | Legal trophy hunts | *NA* |
| **Limpopo National Park** | PA | 10,544 | Inactive | 22 [4] | Decreasing | Decreasing | 4.42 | Human-wildlife conflict & illegal wildlife trade | *NA* |
| **Mahimba Game Farm** | WMA | 349 | Inactive | < 5 [5] | Stable | No mortalities | 0 | No anthropogenic mortalities observed | *NA* |
| **Muanza Game Farm** | WMA | 329 | Inactive | 17 [5] | Stable | No mortalities | 0 | No anthropogenic mortalities observed | *NA* |
| **Niassa Special Reserve** | MUL | 42,353 | Active | 932 (810−1,054) [6] | Decreasing | Increasing | 19 | Poaching | *NA* |
| **Tchuma Tchato Community Programme** | MUL | 38,000 | Active | Unknown | Unknown | No data | No data | No data | *NA* |
| **Zambezi Delta** | MUL | 9,750 | Inactive | 64 [7] | Increasing | Constant | 3.33 | By catch in snares | Lions reintroduced in 2018 after being extirpated |
| **Zinave National Park (Sanctuary)** | PA | 59 | Inactive | 6 [8] | Increasing | No mortalities | 0 | No anthropogenic mortalities observed | Lions naturally recolonized Zinave Sanctuary in 2021 after being extirpated |

(1) Rio Save Safaris Ltd. Unpublished Data. 2022 (Pers Comm), (2) Gorongosa Restoration Project. Unpublished Data. 2022, (3) Sabie Game Park. 2020. Unpublished Report, (4) Everatt, K. T., Kokes, R., Pereira, C. L. 2019. Evidence of a further emerging threat to lion conservation; targeted poaching for body parts. Biological Conservation. 28: 4099–4114, (5) Taylor. Personal Communication. 2023, (6) Niassa Carnivore Project, TRT Conservation Foundation, 2018 Annual Report, (7) Zambeze Delta Conservation. Unpublished Data. 2022, (8) Abrao. Pers Comm. 2024.

national and forest reserves) and wildlife management areas (i.e., hunting concessions, mixed-use areas, community areas, and game farms) comprise approximately 233,249 km² (i.e., 29% of the country; [26]). In particular, we investigate comparatively large and well-monitored resident lion populations (i.e., a demographically cohesive group of lions occupying the same geographic area) occurring in Niassa SR, Gorongosa NP, Tchuma Tchato Community Programme, Zambezi Delta, Coutada 9 and 13 (hereafter, Coutadas 9/13), and Limpopo NP (Fig 1). Lions predominantly occur within these areas, yet some small resident or transient populations are still found outside of this estate [34]. While most lion populations in Mozambique were naturally founded, lions were reintroduced into Coutada 9 in 2010 and the Zambezi Delta in 2018, following post-war functional extirpation [31]. Where lions are considered to be damage-causing animals or are involved in severe cases of HWC, the standing policy of the national wildlife authority (i.e., the Administração Nacional das Áreas de Conservação; ANAC) is to either translocate or euthanize these individuals to mitigate the threat to community lives and livelihoods [34]. Legal trophy hunting quotas for lions are also assigned and regulated by ANAC, with validation by independent partners [46].

## Data collection

We quantified the longitudinal impact of anthropogenic mortality on lion populations in Mozambique using national population estimates and monitoring records between 2010 and 2023. These data were collaboratively complied directly by conservation area managers or research representatives from as many of the Mozambican PAs, wildlife management areas, and mixed-use landscapes as possible. Population size and trend data were derived from intensive monitoring and call-up

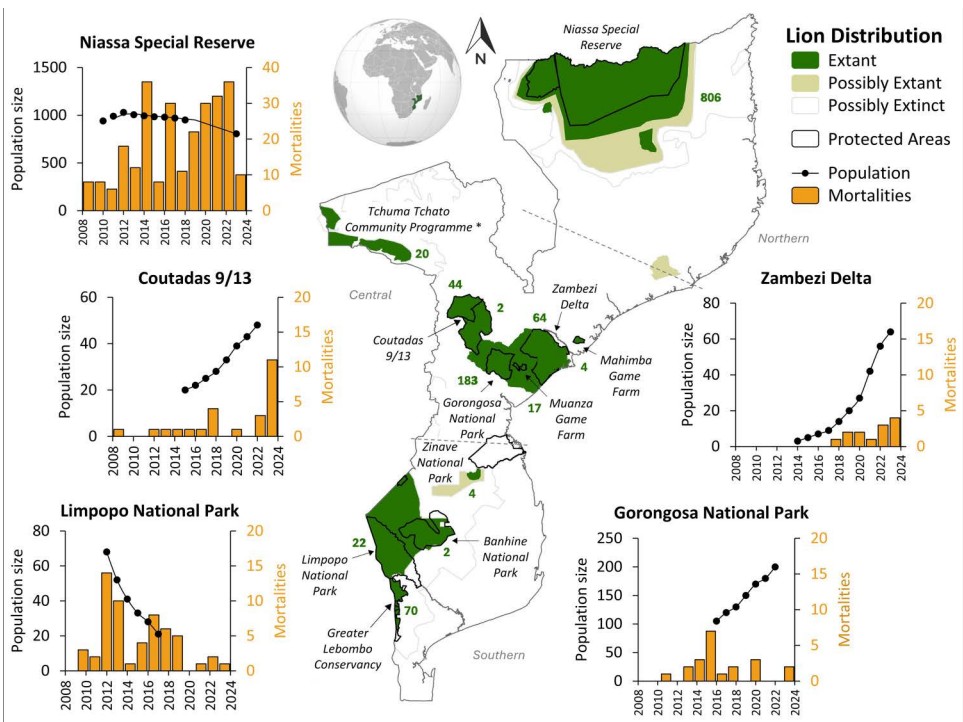

**Fig 1. Resident lion populations in Mozambique, with associated population and anthropogenic mortality trends over time (2010–2023).**
The map shows the current IUCN lion range (extant, dark green; possibly extant, beige; possibly extinct, clear; S1 Appendix in S1 File), and protected areas (grey outlines), indicating known lion populations (labelled) and their most recent population sizes (green numbering). Where longitudinal trend data were available for the 13-year study period, annual lion population size estimates (black circles) and anthropogenically induced mortality incidents (orange bars) are presented. *Data were only available for Chawalo Safari Area as a proxy for the Tchuma Tchato Community Programme.

surveys, while mortality records included the number and demographics of lions killed, cause of mortality (i.e., snared, trapped, poisoned, shot, or vehicular collisions) and, where possible, motivations behind the incident (i.e., accidental, bushmeat by-catch, damage-causing animal, retaliation, targeted poaching, and trophy hunting). Targeted poaching of lions was defined as an anthropogenic mortality with i) no evidence of HWC, ii) evidence of deliberate attempts to kill lions, and iii) body parts removed [38]. Due to the complex nexus of suspected motives, we could not reliably confirm this for all incidents of anthropogenic mortality from the information provided. Incidents where lions were injured but survived following veterinary intervention, were not included in mortality analyses, but were considered in population modelling to demonstrate the value of ongoing intervention efforts.

In addition to population estimates and mortality records, we used information gathered through a range-wide structured questionnaire survey (University of KwaZulu-Natal Human and Social Sciences Research Ethics Committee; HSSREC/00003076/2021; recruitment period: 01-08-2021–31-07-2022), with informed, written consent from participants, to present baseline information on the anthropogenic threats to lions within their Mozambican range, and the resources available to reduce anthropogenic mortalities. Surveys were completed at a subpopulation level, which we define as the non-transboundary lion area that is clearly separated by either political or formal protection boundaries [47]. One completed survey represented one lion subpopulation. We targeted managers of lion subpopulations within formal protected areas, recognised conservancies and game management areas, hunting concessions, and other wildlife areas. A total of 187 experts were invited to participate in the survey, of which 132 responded, yielding a response rate of 71%. Two broad indices were developed from this survey: i) a perceived Anthropogenic Threat Index (ATI) was developed to score the perceived severity of anthropogenically driven threats to lions within a population. From a list of threats, survey respondents were required to indicate whether a particular threat occurred in the area for the past five years and provide a score as to its severity (i.e., ranging from '0', where the threat to lions does not occur in the area, to '4' where the threat is severe enough to potentially result in extirpation). High ATI scores indicated increasingly severe threat intensity, while lower scores (i.e., closer to 0) indicated reduced severity; ii) a perceived Resource and Capacity Index (RCI) was developed in parallel to determine the perceived availability of resources and capacity within PA, wildlife management area, or mixed-use landscape management towards reducing lion mortality. The RCI was based on a series of four trichotomous questions where respondents answered whether area management had sufficient funding and staff, correct anti-poaching gear, and enough vehicles to reduce the illegal killing of lions. For the purpose of this study, we utilized the results of both indices, gathered for Mozambique only, to provide further contextual understanding of threats to lions in Mozambique and the capacity of area managers to reduce the impacts of those threats in their landscapes.

## Statistical analysis

All statistical analyses were run using *R* v4.2.1 statistical software [48]. Spatio-temporal differences in motives and methods behind anthropogenic mortality events were explored through linear or multinomial logistic regression models using the 'multinom' function in the *nnet* package [49]. Where the overall significance of each factor was assessed using Type III ANOVAs implemented in the *car* package [50]. The relative effect of each variable (i.e., motive or method) was plotted by region and year using the *effects* package [51]. Tukey *post-hoc* analyses were then performed in the *lsmeans* package [52] to test for pairwise differences in the relative proportion of each motive or method comparison by region and year.

To evaluate the sustainability of anthropogenic mortality within the local context of Mozambique, we used the individual-based lion population model, which is available as the *pop.lion* package, developed by Loveridge et al. [53]. It is unlikely that all cases of anthropogenic mortality were detected within PAs, wildlife management areas, or mixed-use landscapes, thus we aimed to quantify the approximate rate of detection by comparing model projections based on recorded cases of mortality with recorded population size trends. We ran simulations with five different hypothetical detection probabilities (i.e., 20%, 40%, 60%, 80% and 100%); for example, scenarios that assumed the rate of

detection was 20%, the anthropogenic mortality estimates were multiplied by five for each population, whereafter we visually assessed which scenario aligned most closely with the known lion population size estimates recorded between 2010 and 2023.

We then modelled future projections (i.e., between 2023 and 2040) for each population with varying annual anthropogenic mortality rates (i.e., 0%, 5%, 10%, 15%, 20% and 40%) to provide insight into possible future population trends under the varying levels of proposed mortality. We identified the most likely scenarios based on the recorded trend data corrected by the approximate rate of detection from the first stage of the modelling process (e.g., if the anthropogenic mortality rate based on available data was calculated to be 5% and the rate of detection was estimated to be 60%, estimated anthropogenic mortality would be 8.3%). We calculated the ecological carrying capacity for each lion population to evaluate populations changes relative to these thresholds [54]. All scenarios were run with 1,000 iterations where the mean and quantile range between 0.025 and 0.975 were plotted for each population. For Limpopo NP, we included an additional set of models (i.e., Limpopo NP + K) which accounted for the buffering connectivity effect of lion movement from Kruger NP [38]. This was done by modelling the Limpopo NP and northern Kruger NP as one system and setting Limpopo NP as the edge/sink zone (i.e., the area where offtake occurs [53]). A 17% protected core area was also defined for Niassa SR, leaving 83% 'edge' of the population at modelled risk to anthropogenic mortality. Further details regarding the specific parameters used for each model are provided (S2 Appendix in S1 File).

## Results

### Status, threats and resource availability

We identified 11 conservation areas currently supporting extant lion populations in Mozambique, including a total conservative estimate of approximately 1,208–1,489 lions, covering over 125,970 km$^2$ (i.e., 54% of the available conservation estate across Mozambique). Of these lion populations (Table 1), 37% were reported by conservation managers as currently increasing, 27% were considered stable, 18% were decreasing, and 18% were unknown (Fig 1). Mortality trends for all populations were reported as no mortality (27%), decreasing (9%), constant (28%), increasing (18%) or as having no data (18%) and legal trophy hunting was conducted in approximately 36% of populations across Mozambique.

The perceived ATI revealed the highest threat scores for Coutada 13 and Niassa SR, whereas the Greater Lebombo Conservancy (i.e., Karingani Game Reserve, Sabie Game Park, Massintonto Concession, Mbhatse Concession; Greater Lebombo Conservancy) and Banhine NP reported the lowest threat scores (Fig 2a). In terms of perceived RCI, Niassa SR, Coutadas 9/13, and the Tchuma Tchato Community Programme (i.e., data were only available for Chawalo Safari Area as a proxy for the conservation area) reported the lowest resource and management capacity scores, while Gorongosa NP, Greater Lebombo Conservancy, and the Zambezi Delta reported relatively high scores. After accounting for area and population size, the annual anthropogenic mortality rate of lions per 100 km$^2$ was highest for Muanza Game Farm, followed by Niassa SR, Limpopo NP, and the Greater Lebombo Conservancy (Fig 2b).

### Impact of anthropogenic mortality

A total of 326 anthropogenic mortality incidents (i.e., comprising of 426 individual lions) were recorded across Mozambique between 2010 and 2023 (Fig 3). An additional 16 successful interventions (i.e., comprising of 17 individual lions) were recorded. This constitutes an average of approximately 30 mortalities (i.e., ~2.3% of the total population) and 1.31 interventions (i.e., ~0.1% of the total population) per year (at a mean of 5.77 ± 2.67 [SE] lions per population). There was a significant increase in the number of annual anthropogenic mortalities observed ($F = 6.99$, df = 1 and 12, $P < 0.05$), increasing from 9 to 49 mortalities (i.e., 1–4% of the total population) over the 13-year period. Since 2012, the number of anthropogenic mortalities has exceeded 26 individuals each year, with the highest numbers recorded in 2015 ($n = 49$), 2017 ($n = 48$), 2020 ($n = 41$), and 2022 ($n = 49$). Demographically, males (68%) were killed more frequently than females

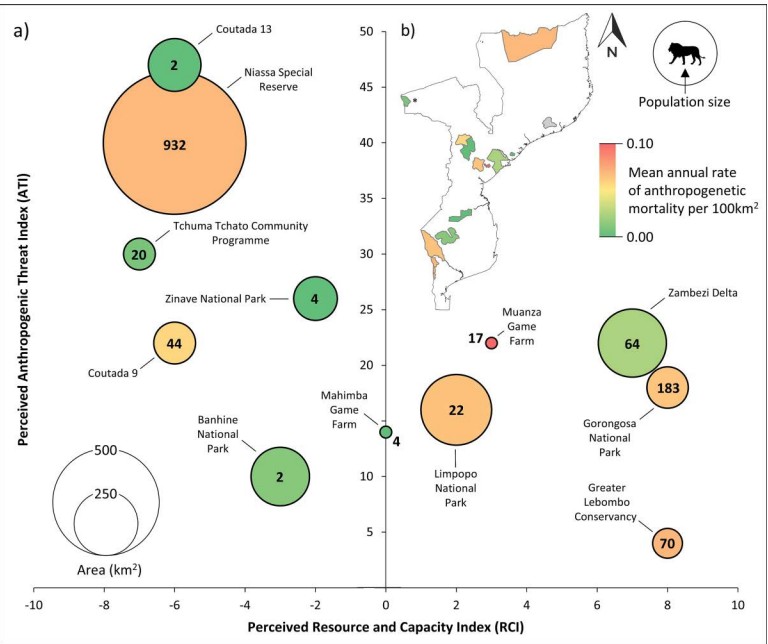

**Fig 2. Lion population size by area-based anthropogenic mortality rate relative to perceived threat, resource and management capacity indices across Mozambique.** a) Lion population size (bold number inside bubbles) for each area (bubble size) and b) mean annual rate of anthropogenic mortality per 100km² between 2010 and 2023 (green-yellow-red colour gradient) are indicated relative to perceived anthropogenic threat index (ATI), and resource and capacity index (RCI). ATI is the perceived severity of anthropogenically-driven threats to lions within a population, with higher index values indicating higher anthropogenic threats. RCI is the perceived availability of resources to management for reducing lion mortality, with higher values indicating greater resource and management capacity.

(32%) based on known sexes, while 83% of mortalities were adults, followed by subadults (14%), and cubs (3%) based on known ages.

The greatest proximate causes of lion mortalities were illegal motives (i.e., 65% of incidents), comprising bushmeat bycatch (27%), targeted poaching for parts (25%), and retaliatory killing (13%), followed by legal trophy hunting (33%). There was a significant difference in motive for anthropogenic mortality over time ($F=30.065$, df=6, $P<0.001$; S1 Table in S1 File), with the prevalence of bushmeat bycatch and targeted poaching increasing, and trophy hunting decreasing over time (Figs 3–4). There was also a significant difference in the method used for lion killings over time ($F=152.97$, df=65, $P<0.001$; S1 Table in S1 File), with the most prominent changes being an increase in poisoning and decrease in shooting of lions over time. There were significant differences in motives for lion killings across regions ($F=226.22$, df=18, $P<0.001$). Lion mortalities in the southern region comprised largely of targeted poaching and retaliatory killing, while bushmeat bycatch predominated the central region, and the northern region was mostly affected by trophy hunting and targeted poaching (S1 Table in S1 File). There were also significant differences in methods used between regions ($F=217.21$, df=15, $P<0.001$), with poisoning and snaring used in the south, gin trapping and snaring in the central region, and trophy shooting in the north.

Targeted poaching of lions for body parts increased significantly over time ($F=5.22$, df=1 and 10, $P<0.05$), with an average of one lion killed per year between 2010 and 2017, to an average of 7 lions per year between 2018 and 2023 (Fig 5). While illegal wildlife trade related poaching was evident in all populations, most incidents of body part removal were reported in the northern (e.g., skins, paws, and whole bodies) and southern (e.g., paws, face/head/skulls, and flesh/meat) regions.

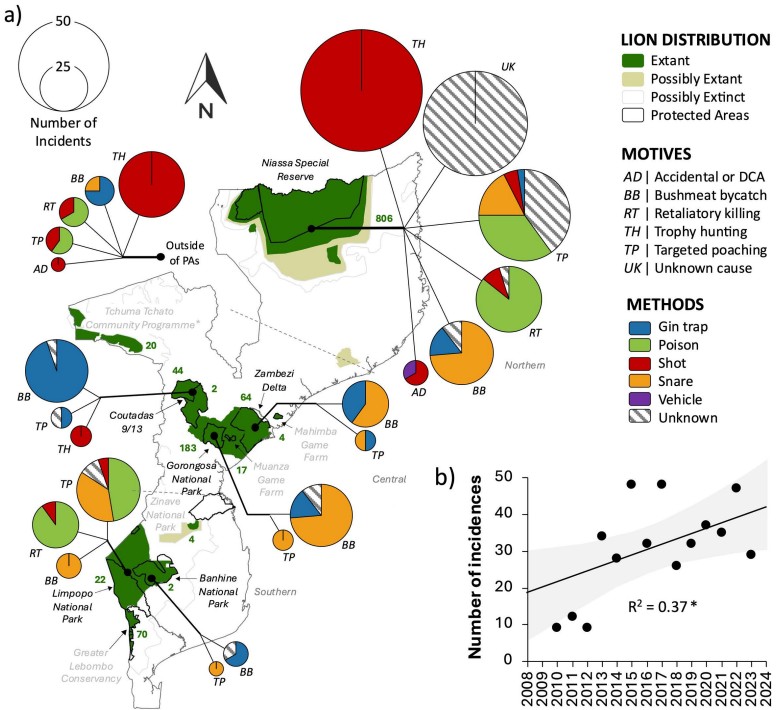

**Fig 3. Typology of anthropogenic lion mortality across Mozambique between 2010 and 2023.** Indicated are the typological motives (i.e., AD – accidental or damage-causing animal, BB – bushmeat bycatch, RT – retaliatory killing, TH – trophy hunting, TP – targeted poaching, and UK – unknown cause) and methods (i.e., blue – gin trap, green – poison, red – shot, orange – snare, purple – vehicle, and grey hashing – unknown) of anthropogenic mortalities. **a)** The map shows the current IUCN lion range (extant, dark green; possibly extant, beige; possibly extinct, clear; S1 Appendix in S1 File), and protected areas (grey outlines), indicating known lion populations (labelled) and their most recent population sizes (green numbering). The circle size indicates the number of incidents by reported motive and method. Indicated also are the b) significantly increasing trend of anthropogenic mortality incidents over time.

## Population viability modelling

Visual assessment of the population projections for each area under varying detectability levels of anthropogenic mortality (Fig 6) showed that Gorongosa NP, Coutadas 9/13 and the Zambezi Delta aligned best with population size trends when detection was set around 100%. This validation of near-complete detection indicates that a relatively greater degree of confidence may be assigned to projections for these populations. For Niassa SR, the scenario with detection rate set at 20% aligned most closely with the observed population trend, while all larger detection rates produced projections that far exceeded population estimates between 2010 and 2023. The 20% detection rate scenario was also the best fit for Limpopo NP when the buffering effect of Kruger NP was included, while the scenario with a slightly higher detection rate of 40% was the closest fit for Limpopo NP when Kruger NP was excluded.

The annual anthropogenic mortality rate in Gorongosa NP is projected to remain low and stable at approximately 1.5%, with ongoing interventions likely further reducing this rate (Fig 7). Coutadas 9/13 and the Zambezi Delta are projected to remain low to moderate with anthropogenic mortality rates at approximately 9% and 8% respectively, although current levels of intervention appear to reduce these rates by 2–3%. In Niassa SR, the annual anthropogenic mortality rate outside the well-protected core area is projected at approximately 3.2%, however, given the relatively low detection rate (i.e., 20%), the true annual anthropogenic mortality rate for Niassa SR is likely to be much higher at around 16%. In Limpopo NP, the annual anthropogenic mortality rate is projected as being low to moderate at approximately 8%, however, as the detection rate is likely very low (i.e., 20–40%), the actual rate of annual anthropogenic mortality could be high at 19.8% (i.e., excluding Kruger NP) to extremely high at approximately 40% when accounting for connectivity with Kruger NP.

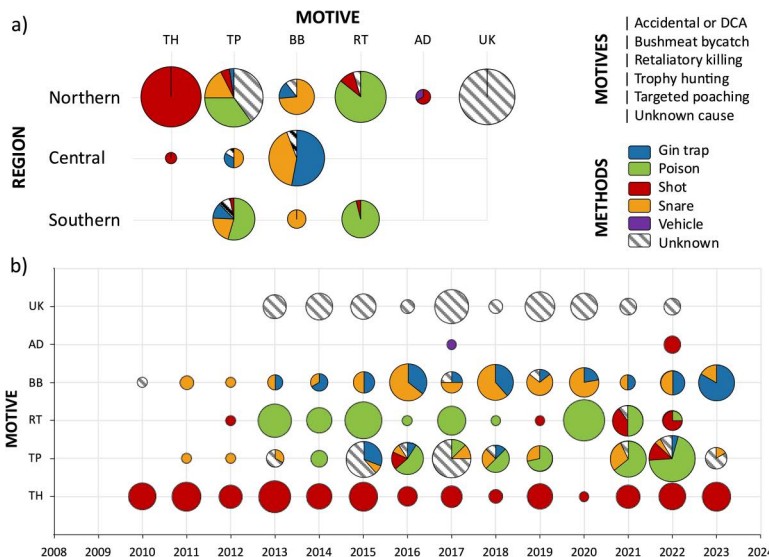

**Fig 4. Typology of anthropogenic lion mortality summarized by a) region and b) over time in Mozambique between 2010 and 2023.** Indicated are the typological motives (i.e., AD – accidental or damage-causing animal, BB – bushmeat bycatch, RT – retaliatory killing, TH – trophy hunting, TP – targeted poaching, and UK – unknown cause) and methods (i.e., blue – gin trap, green – poison, red – shot, orange – snare, purple – vehicle, and grey hashing – unknown) of anthropogenic mortalities.

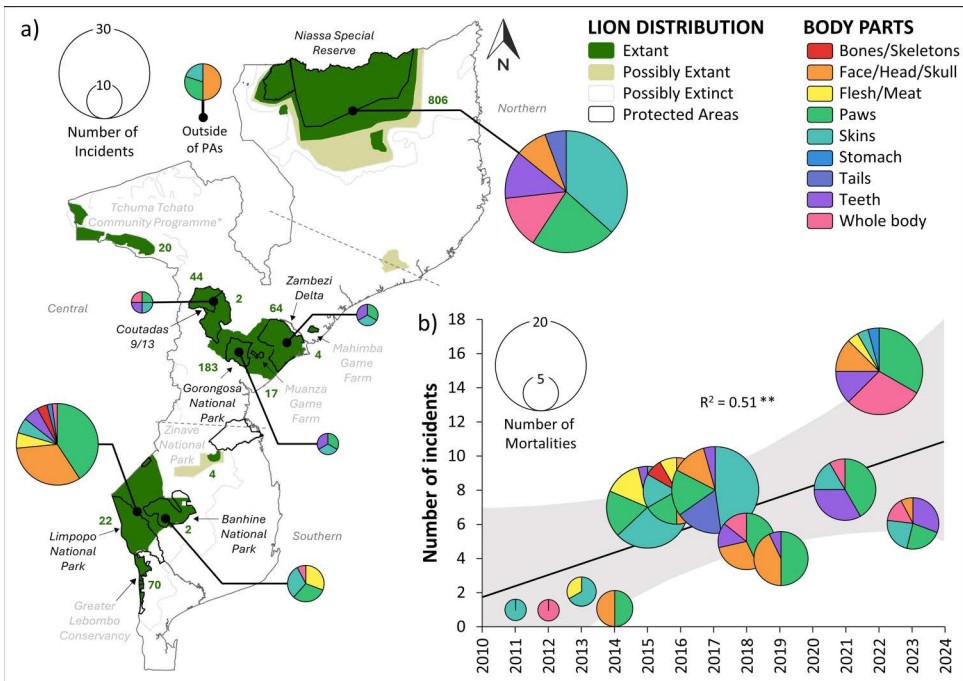

**Fig 5. Lion body parts removed for illegal wildlife trade across Mozambique 2010–2023).** The map shows the current IUCN lion range (extant, dark green; possibly extant, beige; possibly extinct, clear; S1 Appendix in S1 File), and protected areas (grey outlines), indicating known lion populations (labelled) and their most recent population sizes (green numbering). Indicated are the number of confirmed incidents (circle size) of anthropogenically driven lion mortality from which body parts were taken and presumably intended for the illegal wildlife trade, and the proportion of body parts (red – bones/skeletons, orange – face/head/skull, yellow – flesh/meat, green – paws, aquamarine – skins, blue – stomach, purple – tails, violet – teeth, and pink – whole body) taken by a) area and b) year.

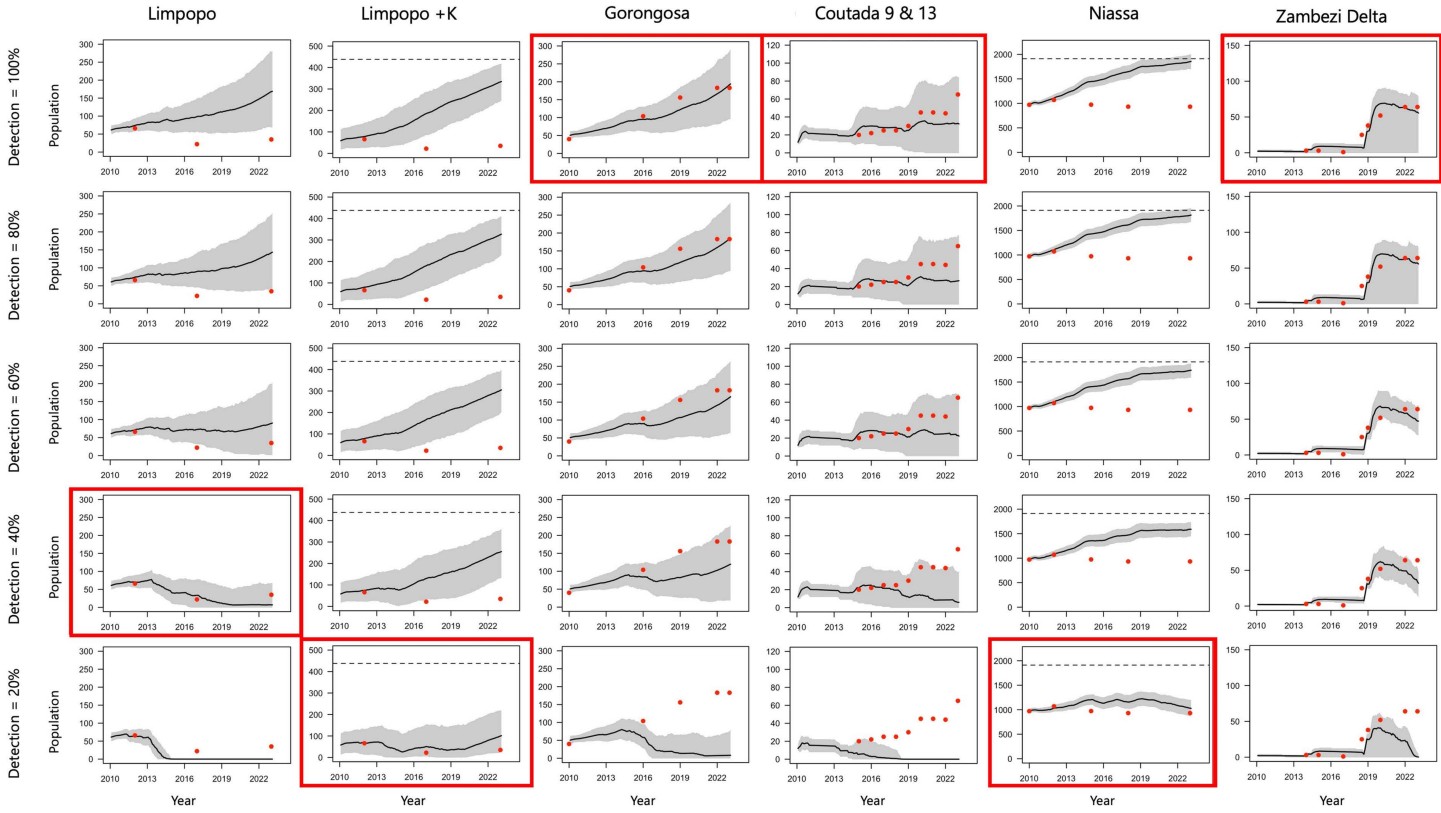

**Fig 6. Lion population projections in Mozambique based on recorded levels of anthropogenic mortality with varying levels of assumed mortality detection.** Plots outlined with red squares indicate the most likely scenarios based on visual assessment of model fit (black line with 95% CIs in grey). Red dots represent annual population estimates for each lion population and horizontal dashed lines indicate estimated ecological carrying capacity. The K (Kruger National Park) associated with Limpopo National Park represents the model run with the inclusion (+) of Kruger NP as a buffering effect to Limpopo NP, given the connectivity between these two PAs.

Future simulations indicate that for all populations, eliminating anthropogenic mortality would lead to natural population growth such that by 2040 these populations would approach ecological carrying capacities. Gorongosa NP was projected as having the highest annual population growth rate of 6.5% and is projected to reach its ecological carrying capacity by 2040. The Zambezi Delta and Coutadas 9/13 support projected growth rates of approximately 0.8% and 0.4% per annum respectively and are not likely to achieve their ecological carrying capacities by 2040, although these populations are projected to display increasing trends if intervention capacity is maintained. The annual population growth rate projected for Niassa SR is −0.2%, indicating a population at stability tending towards decline, where projected levels of anthropogenic mortality are likely to keep suppressing the population at around half of its ecological carrying capacity by 2040. In Limpopo NP, the projected annual population growth rate is −0.1% to −0.4%, with and without the buffering effect of Kruger NP respectively, indicating that without this effect, the population will likely be extirpated by 2040.

To secure projected stable or growing annual population growth rates, given varying degrees of confidence in detection, annual anthropogenic mortality targets should be maintained at ≤5% for Gorongosa NP, whereas the Zambezi Delta and Coutadas 9/13 should set their annual anthropogenic loss targets at ≤5% (S2 Table in S1 File). Niassa SR should aim to keep annual anthropogenic mortality at ≤ 10% outside of the well-protected core area. In contrast, the Limpopo NP should take drastic steps to reduce annual anthropogenic mortality rates by half and keep these at ≤5% (without Kruger NP) or at ≤20% (with Kruger NP) to remain stable and viable or support recovery by 2040.

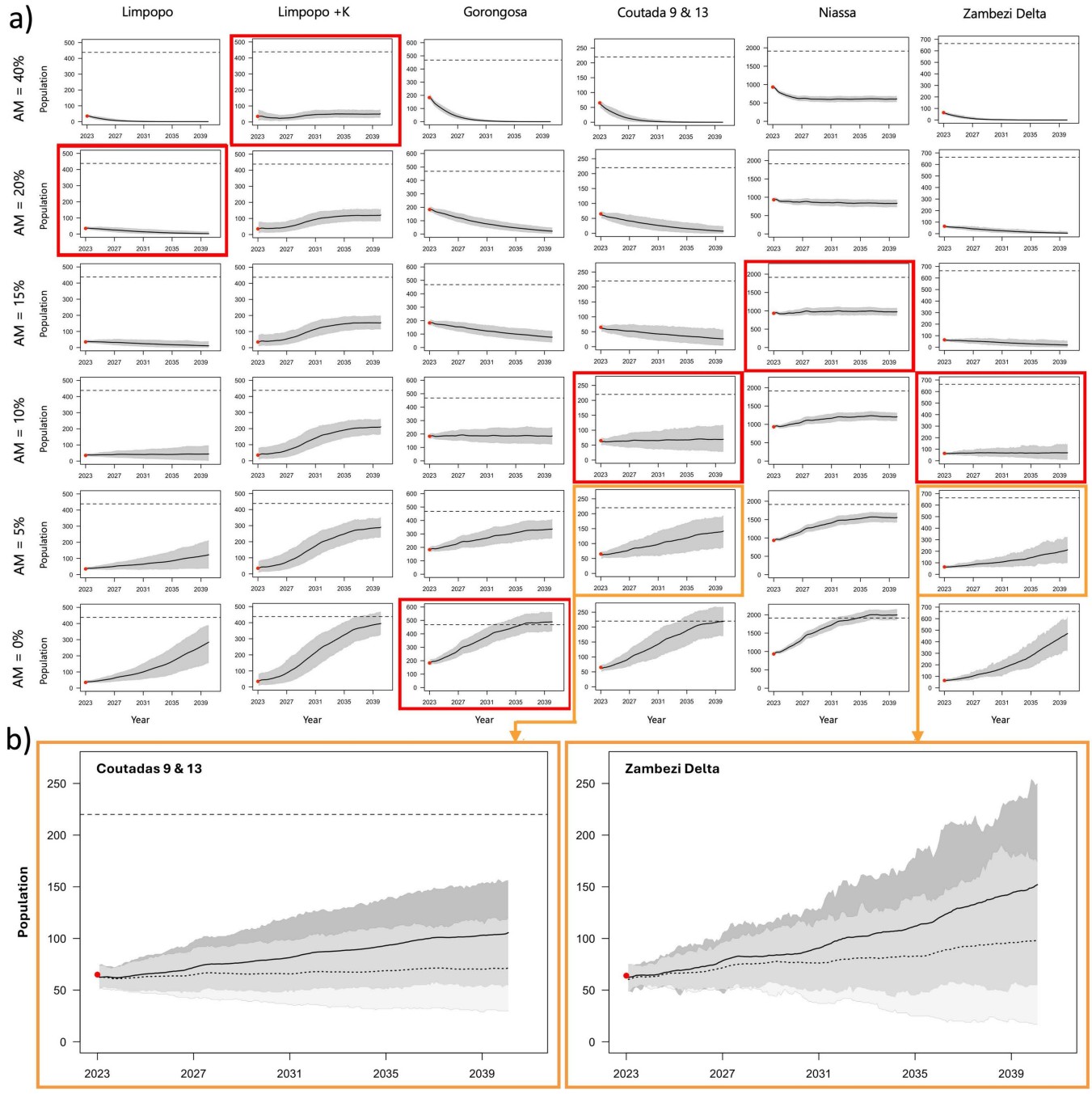

**Fig 7. Lion population projections for in Mozambique under varying rates of anthropogenic mortality and management intervention. a)** Plots with red squares indicate the most probable outlook based on current anthropogenic mortality rates corrected by estimated rates of detection (black line with 95% CIs in grey). Horizontal dashed lines indicate ecological carrying capacity as calculated for each of the populations. **b)** Plots with orange squares indicate the most probable outlook based on current anthropogenic mortality rates corrected by estimated rates of detection and the reported degree of intervention.

## Discussion

Anthropogenic mortality has driven large carnivore population declines and extirpations across their range [12]. The inherent resilience of these populations is generally a function of population size and landscape integrity, but also resource and management capacity (i.e., security, monitoring, and funding) relative to the scale of these threats (i.e., coexistent human density, socio-economic dependencies, and tolerance; [12,23,55]). Quantification of anthropogenic mortality throughout post-war Mozambique revealed that bushmeat poaching, among other threats, is a significant suppressor of regional lion population recovery. Predictive modelling based on recent population trends, considering relative anthropogenic mortality rates, suggests that most resident lion populations in Mozambique will remain suppressed below ecological carrying capacity or suffer further declines over the next 15–20 years. These trends are further compounded in comparatively large populations (e.g., Niassa SR and Limpopo NP), as anthropogenic mortality remains under-detected and there is arguably an unstable dependency on well-protected core areas or buffering populations. In contrast, management capacity in comparatively smaller populations (e.g., Gorongosa NP and Zambezi Delta), improves detection and effective mitigation of anthropogenic mortality; but the long-term sustainability of these models will be increasingly challenged by growing lion populations. Moreover, the ubiquity and significant increase of targeted poaching of lions to supply IWT in body parts [38,39], itself often intertwined with bushmeat poaching and retaliatory killing [56], presents a rapidly growing threat. Our documented levels of illegal anthropogenic mortality (i.e., retaliatory persecution, as well as bushmeat and targeted poaching for body parts), exacerbated in smaller populations with limited management capacity, and populations with high transfrontier threats for illegal killing of lions for parts (i.e., Niassa SR and Limpopo NP), threaten the long-term viability of Mozambique's lion populations.

Of the 426 anthropogenically-driven lion mortalities recorded between 2010–2023, 65% of incidents were of illegal causes (i.e., bushmeat bycatch, as well as targeted and retaliatory killing), reiterating concerns raised over largely cryptic and unregulated threats to lions throughout Mozambique. In Niassa SR, which supports 62–77% of the national lion population, anthropogenic mortality rates range from low to high (3.2–16%), with the precision of these estimates constrained by low detection rates (~20%). While current efforts appear sufficient to maintain this population size at approximately half of its ecological capacity, any increase in the annual anthropogenic mortality rate (i.e., 15–20%) or fragmentation of the core area is likely to precipitate future population decline [46]. Niassa SR management should aim to increase overall detection and reduce these mortality rates to ≤10% to ensure population growth and resilience. With near complete detection, Gorongosa NP represents confident post-war lion population recovery, with projections approaching ecological capacity by 2040 given the lowest annual anthropogenic mortality rate (2%). Such site security and restoration is likely a result of the comparatively small area, intensive lion-specific monitoring efforts, and anti-poaching coverage facilitated by long-term investment through public-private partnerships [30,57,58]. As both Coutadas 9/13 and Zambezi Delta populations are reintroduced into relatively large areas, their growth is substantial, but will likely not attain ecological capacity within the projection timeframe. However, given increasing annual anthropogenic mortality rates in both areas, populations may begin to stabilize or decline. Management in these areas should maintain overall detection and reduce these mortality rates to ≤5%, especially as these populations continue to grow and their exposure to illegal anthropogenic threats intensifies [31]. Of greatest concern, however, is that predictive models, at low detection rates (~40%), suggest that without the buffering effect of Kruger NP, the lion population in Limpopo NP will likely be functionally extirpated by 2030, a compounding result of a relatively small population size and unsustainable levels of anthropogenic mortality [29,38]. This collective status and projection of lion populations in Mozambique highlights the severity of the monitoring and enforcement crisis currently threatening these and other regional strongholds [24,38,40,47].

Monitoring population dynamics and threats are challenging for elusive carnivores in expansive landscapes, and predicting long-term trends pose several caveats and limitations [43,59,60]. Firstly, perceived anthropogenic threat indices are dependent on population persistence and may be lower than reality if these populations are reduced or extirpated (e.g., Banhine NP and Limpopo NP). Secondly, while the significant increase in anthropogenic mortality identified in this

study may be an artefact of increased monitoring efforts, these findings corroborate growing evidence of increased legal and illegal trade in lion body parts regionally [35,36,40,61]. Thirdly, while more robust methods for estimating lion population size are increasingly recommended [43,44], consistent application of traditional methods provided sufficient insights into longitudinal population trends [18]. Fourthly, efforts to monitor lion mortality and indicators of specific threats (e.g., lion conflict events, and anti-poaching coverage) vary substantially across landscapes and depend on various factors such as PA size and accessibility, as well as monitoring capacity and efficacy [13,62], resulting in variable detection rates and therefore confidence in projections [7,25,40,55]. Finally, while the number of reported lion mortalities has fluctuated over time, some of this variability may be attributed to seasonal accessibility or temporal variability in retaliatory killing, and demand for bushmeat or lion body parts for trade [13,62]. In addition, the detection of lion mortality may also be masked by cryptic poaching (i.e., poachers concealing their activities and avoiding detection) in response to increased antipoaching efforts [63]. In contrast, legal trophy hunting is comparatively well documented and regulated, thus these lion mortalities are likely near complete records [46]. Nevertheless, such regional population trend modelling accounts for such variability and provides important insights into the impact of this pervasive threat to lions across Mozambique.

That Mozambique is vulnerable to bushmeat poaching and increasingly threatened by IWT, is not unique, as many lion range States have experienced turbulent political histories with ongoing socioeconomic fragility which challenge conservation efforts [14,18,27]. However, there are many factors predisposing Mozambique to these threats, most of which are driven by protracted multi-dimensional poverty, that continues to suppress large carnivore populations [29,31]. The rapidly growing human population continues to exert increasing pressure on natural resources through further encroachment into conservation areas [32,64,65], deforestation for logging and agriculture, and bushmeat poaching to fuel consumptive and commercial trade demands [29,35,66]. Relatively weak and poorly implemented penal systems, inadequate and corrupt law enforcement, as well as a lack of wildlife-based benefits and opportunities for local enterprise are also highlighted as potential drivers of bushmeat poaching [14,67]. Mozambique is geographically vulnerable with an exposed coastline which is generally poorly monitored, and long borders with neighboring countries who experience similar transnational wildlife crime [68]. Furthermore, the susceptibility of lion populations to anthropogenic mortality is likely exacerbated by existing poaching syndicates, and wide-reaching criminal economies, with accessibility to international ports and trade routes [37,68]. For instance, between 2012 and 2017, Everatt et al. [38] estimated a 66% lion population decline in Limpopo NP, with targeted lion poaching contributing to 61% of anthropogenic mortalities. This declining population trend was matched by an increasing demand for lion body parts in the IWT, particularly from 2014 [38], alongside a near five-fold parallel increase in legal exports of lion skeletons from southern Africa to south-east Asia [36,61]. This is likely evidence of expanding trader and consumer markets, with lions increasingly used to supplement previously targeted species, especially tigers (*Panthera tigris*), which have become less accessible due to population declines and improved protection efforts in Asia [35]. Though such informed speculation should not be conflated with interpretation [56]. Such Asian influence in regional lion body part trade is not limited to end-user demand, rather, there is a chain of local exploitation and influence which feeds into syndicated international IWT, that is particularly pronounced in Mozambique [35,38]. Similar trends have been recorded for Niassa SR since 2015 [39]. This largest PA in the country, remains under-resourced resulting in a poor detection of anthropogenic mortalities and is at the projected threshold of unsustainable offtake. Comparably, Limpopo NP has greater detection, though this is likely an artefact of a severely compromised lion population [38] which shows a significant transnational source dependency on Kruger NP to avoid functional extirpation [38]. In an effort to combat wildlife-related crime, Mozambique enacted the Conservation Areas Law (Act N°7 of 2017), which amends and strengthens provisions from Act 16 of 2014. This new legislation aligns with international conservation standards and imposes more stringent penalties for wildlife crimes, particularly those involving species, like lions, that are listed under the Convention on International Trade in Endangered Species of Wild Fauna and Flora, with prison sentences up to 12 years. However, law enforcement in the field and consistent application of the law in the courts remains problematic [68]. Despite a quarter of the country being formally designated for conservation [69],

only ~16% of the total funding required for effective PA management is available, making Mozambique one of the most under-funded countries for lion conservation [23].

The importance of Mozambique among lion range States in Africa cannot be neglected, and yet these resident populations are under significant pressure, requiring urgent national-scale action. To address these threats, we encourage improved monitoring of lion populations, allowing for greater responsiveness when problems arise and providing a stronger basis for evidence-based quota setting. Additionally, improved detection of anthropogenic mortality, through standardized monitoring systems, would help to generate more accurate data to assess impacts on lion population trends and evaluate the efficacy of current conservation interventions [40,70]. Site security remains an ongoing issue and coordinating anti-poaching operations more effectively may reduce pressure on lion populations locally, while developing integrated inter-agency task forces to coordinate joint patrol operations and review investigations may improve regional security [63]. Tackling bushmeat poaching and IWT requires significant efforts from local and national authorities towards identifying weak points for targeted investigations to disrupt IWT networks [71]. Direct intervention with judiciary systems through sensitization of magistrates is required to ensure a robust understanding of the impact of wildlife crimes, as well as courtroom monitoring to flag inconsistencies in the application of the law [14]. Furthermore, appropriate zonation of PAs is required on a national scale through more accurate delineation, and enforcement of these policies to prevent further inward immigration [27,28]. Trophy hunting represents an important land-use practice in Mozambique [69], funding approximately 30% of the annual operational costs for Niassa SR, along with anti-poaching and management activities, while supporting local communities and government initiatives [72]. While we are cautious in recommending reduced legal offtake quotas, especially given that legal offtake is comparatively well-monitored and detected in Niassa SR [46], compensating for current illegal anthropogenic mortalities in quota setting, even temporarily, may benefit local population recovery [31]. Refining opportunities for alternative modes of wildlife-related investment opportunities in wildlife management areas and mixed-use landscapes, as well as developing tripartite partnerships between government, hunting operators and NGOs can help bridge the gap between revenues from hunting and funds needed for effective management. Mozambique must attract more inward investment into the conservation space, although, this is dependent on ANAC creating a long-term enabling environment for potential investors by developing and adhering to clear policy on public-private partnerships, including simplifying bureaucracy and processes for engaging in the country. The systemically low resource capacity and high threat levels across these large PAs requires a significant upscaling of co-management intensity which will rely on external operational and financial support [23]. However, given the underlying socio-economic fragility of many rural landscapes alongside PAs, it is essential that these actions and funding models be built around the needs and opportunities of impacted communities, such that they become enfranchised stakeholders in lion conservation with clear incentives to coexist.

## Supporting information

**S1 File. S1 Appendix.** Descriptions of lion populations in Mozambique. **S2 Appendix.** Population modelling details. **S1 Table.** Tukey post-hoc test results derived from the multinomial linear regression models. **S2 Table.** Details of estimated detection rates, anthropogenic mortality rates (AMR) and recovery targets for lion populations within Mozambique. Included are current AMRs, trends in AMRs, and targets for AMR reduction needed to promote lion recovery. Effects of interventions on AMRs are also provided to highlight the importance of monitoring and veterinary capacity. (DOCX)

## Acknowledgments

We would like to thank all researchers and organizations who provided data for this study. The authors are also grateful to the Administração Nacional das Áreas de Conservação (ANAC) for approving and supporting this work.

## Author contributions

**Conceptualization:** Willem D. Briers-Louw, Colleen Begg, Vincent N. Naude, Samantha K. Nicholson.

**Data curation:** Samantha K. Nicholson.

**Formal analysis:** Willem D. Briers-Louw, Andrew Loveridge, Matthew Wijers, Vincent N. Naude, Samantha K. Nicholson.

**Investigation:** Willem D. Briers-Louw, Vincent N. Naude, Samantha K. Nicholson.

**Methodology:** Willem D. Briers-Louw, Andrew Loveridge, Matthew Wijers, Vincent N. Naude, Samantha K. Nicholson.

**Project administration:** Willem D. Briers-Louw, Vincent N. Naude, Samantha K. Nicholson.

**Supervision:** Vincent N. Naude, Samantha K. Nicholson.

**Validation:** Willem D. Briers-Louw, Vincent N. Naude, Samantha K. Nicholson.

**Visualization:** Willem D. Briers-Louw, Vincent N. Naude, Samantha K. Nicholson.

**Writing – original draft:** Willem D. Briers-Louw, Matthew Wijers, Vincent N. Naude, Samantha K. Nicholson.

**Writing – review & editing:** João Almeida, Willem D. Briers-Louw, Agostinho Jorge, Colleen Begg, Marnus Roodbol, Hans Bauer, Andrew Loveridge, Matthew Wijers, Rob Slotow, Peter Lindsey, Kristoffer Everatt, Holly Rosier, Sean Nazerali, Lizanne Roxburgh, Hugo Pereira, Mercia da Conceicao, Armindo Araman, Osvaldo J. Abrao, Alison J. Leslie, Franziska Steinbruch, Vincent N. Naude, Samantha K. Nicholson.

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
