## [Decision Letter · Decision Letter 0]

2 May 2025

PONE-D-25-03127Unsustainable anthropogenic mortality threatens the long-term viability of lion populations in MozambiquePLOS ONE

Dear Dr.  Briers-Louw,

Thank you for submitting your manuscript to PLOS ONE. After careful consideration, we feel that it has merit but does not fully meet PLOS ONE’s publication criteria as it currently stands. Therefore, we invite you to submit a revised version of the manuscript that addresses the points raised during the review process.

Dear Authors,I had the pleasure to read you manuscript and I think it is very interesting and data deserve to be shared with the public. I think it is important to have a better view of the conservation status of the lion in Africa and specifically in Mozambique and your manuscript could add an important contribution.One of the weakest point is that you included so many information that sometimes it is not easy to understand them for a broader readership. I suggest to shorten some parts as suggested by reviewer and try to be more focused on the hypothesys.Please carefully follow reviewers suggestions.

We look forward to receiving your revised manuscript.

Kind regards,

Francesco Bisi, Ph.D.

Academic Editor

PLOS ONE

Journal Requirements:

2. We note that your Data Availability Statement is currently as follows: All relevant data are within the manuscript and in Supporting Information files.

4. We note that Figures 1-4 in your submission contain map/satellite images which may be copyrighted. All PLOS content is published under the Creative Commons Attribution License (CC BY 4.0), which means that the manuscript, images, and Supporting Information files will be freely available online, and any third party is permitted to access, download, copy, distribute, and use these materials in any way, even commercially, with proper attribution. For these reasons, we cannot publish previously copyrighted maps or satellite images created using proprietary data, such as Google software (Google Maps, Street View, and Earth). For more information, see our copyright guidelines: http://journals.plos.org/plosone/s/licenses-and-copyright.

 1. You may seek permission from the original copyright holder of Figures 1-4 to publish the content specifically under the CC BY 4.0 license. 

Additional Editor Comments:

To help a broader readership understand the problem, I think it would be helpful to clarify how you define distinct lion populations. Is the term population appropriate in all cases? For example, when there are only a few individuals, such as the two lions shown in Figure 1, can that be considered a population, or would pride be a more accurate term?

In Supplementary Material 1, you describe how the population in Coutadas 9/13 was extirpated, followed by reintroduction efforts, and that a hunting quota of two lions per year has been established. However, Figure 1 reports a population estimate of only two lions.

It would be important to also describe the structure of the survey used to generate the ATI and RCI indices, including at least the number of responses collected.

As you mention, data on lion mortalities are incomplete, since not all deaths are detectable. I'm curious whether the reporting effort remained constant throughout the study period, or if it might have varied due to awareness campaigns or other dedicated projects.

Reviewers' comments:

Reviewer's Responses to Questions

**Comments to the Author**

1. Is the manuscript technically sound, and do the data support the conclusions?

Reviewer #1: Yes

Reviewer #2: Yes

2. Has the statistical analysis been performed appropriately and rigorously? 

Reviewer #1: Yes

Reviewer #2: Yes

3. Have the authors made all data underlying the findings in their manuscript fully available?

Reviewer #1: Yes

Reviewer #2: Yes

4. Is the manuscript presented in an intelligible fashion and written in standard English?

Reviewer #1: Yes

Reviewer #2: Yes

5. Review Comments to the Author

Reviewer #1: The authors sought to measure the long-term effects of anthropogenic mortality on Mozambican lion populations in their work "Unsustainable anthropogenic mortality threatens the long-term viability of lion populations in Mozambique." The conclusions derived from this well-structured manuscript are consistent with the findings. I aim to offer some very helpful recommendations to raise the general level of clarity in your research and the caliber of your analysis. I believe you will be able to handle my recommendations, and I think they seem doable to you.

Lines 62 – 119: Please reduce this part of the manuscript.

Lines 120 – 129: Please explain in detail your hypothesis and predictions. You need to expand this section if you would want to express exactly what you want to do.

Lines 196 – 227: The authors must add all the R codes used in this study in the supplementary materials. Please add R codes well described and understandable.

Line 343: Please DELETE all the references from the results section.

Lines 361 - 362: I think that you should add these two important references as examples to support your sentence: “Anthropogenic mortality has driven large carnivore population declines and extirpations across their ranges.”. I would like to suggest:

Ahmad, F., et al., (2025). Asiatic black bear in Pakistan: a comprehensive review and conservation indications. Mammalian Biology. https://doi.org/10.1007/s42991-025-00479-x

Lohr, M. T., et al., (2025). Widespread detection of second generation anticoagulant rodenticides in Australian native marsupial carnivores. Science of the Total Environment, 967(17883), 178832.

Lines 365 – 508: This part of the manuscript should be reduced.

Figures 1 - 4: Please add the north symbol and the scale in the map.

Reviewer #2: Comments to the authors on manuscript titled “Unsustainable anthropogenic mortality threatens the long-term viability of lion populations in Mozambique”

I thoroughly enjoyed reading this manuscript. It is a compelling, well-written, and a highly relevant contribution to lion conservation. The authors present a clear and rigorous analysis of a pressing issue, and the integration of field data with modelling provides valuable insights for conservation policy and practice across Mozambique and beyond. The attention to regional variation and the nuanced discussion of anthropogenic pressures are particularly noteworthy. That said, the manuscript is very long and could benefit from being made more concise. Streamlining certain sections—particularly the Introduction and Discussion—would improve readability and sharpen the paper’s overall impact without compromising the quality of the work.

Apart from reducing the length I only have minor comments to address, mostly to improve grammar and flow.

Line 54 and 55: Remove the repeated use of “and” inserting commas instead, just keep ”and” for the end statement “and human-wildlife conflict”.

Line 65: Spell out IUCN on its first mention

Line 82: The word gazetted does not seem to fit well here, perhaps replace with a different word

Line 85: break this statement into two sentences as it is very long. I suggest putting a full stop after “resident communities”.

Lines 97-100: Consider adding a statement explaining why lions are being targeted for their body parts, for readers who may be unaware of why this trade exists.

Line 133: Change “gazetted” to Prioritized or something similar

Line 161: Just state that you “quantified the longitudinal impact.....” . There should not be aims mentioned in the methods, this kind of statement is suited for the introduction.

Lines 175-194. There are a lot of acronyms added in this section, and throughout the paper in general. While acronyms are useful for cutting down repeated words, in some cases it becomes confusing for the reader to keep track. I suggest cutting back on some of the acronyms, particularly in this section.

Line 196: Remove the word “the” after differences

Line 234-235: This statement is a little confusing, do you mean a daily average, an annual average or just the total average across the entire time frame?

Line 254: Change “towards” to “for”, it reads better.

Line 269: Also state what DCA is, again too many acronyms to keep track of.

Line 379: Can you provide any more information (perhaps this is better suited in the methods) as to how and when agencies could assign mortalities to targeted poaching for the IWT rather just opportunistic when lions were killed for other reasons? I realize how inherently difficult this might be, but is it just the case then when body parts are removed it is always assigned to poaching for IWT?

Line 386-389: This sentence is quite the mouthful and does not flow or read well, please revise.

Line 400: Insert “of” before “Limpopo

Lines 454-456: Do we know what is driving this increase in demand? Is the buyers market growing or are perhaps other previously sought after species for the IWT more difficult to obtain leading to a shift towards lions? Perhaps you add a statement and reference to other studies if anything is known about changes to this market

Line 467: I think you should add a statement to explain what this new legislation means for lion conservation in the country

Lines 474-478: This sentence is too long, break it up into two statements to improve flow.

Line 499: Inset “on” before ANAC

Figures

Figure 3: This figure is too chaotic and difficult for a reader to easily understand what is going on. I suggest this is changed to display the information in a more reader friendly format, even if this means breaking it up into a few smaller figures.

Figure 5 & 6: Are quite small and the legends seem a little bit blurry, please revise.

6. PLOS authors have the option to publish the peer review history of their article (what does this mean? ). If published, this will include your full peer review and any attached files.

**Do you want your identity to be public for this peer review?** For information about this choice, including consent withdrawal, please see our Privacy Policy .

Reviewer #1: No

Reviewer #2: No

---

## [Author Response · Author response to Decision Letter 1]

12 May 2025

See attached "Response to Reviewers" file.

---

## [Editor Report · Decision Letter 1]

18 May 2025

Unsustainable anthropogenic mortality threatens the long-term viability of lion populations in Mozambique

PONE-D-25-03127R1

Dear Dr. Briers-Louw,

We’re pleased to inform you that your manuscript has been judged scientifically suitable for publication and will be formally accepted for publication once it meets all outstanding technical requirements.

Kind regards,

Francesco Bisi, Ph.D.

Academic Editor

PLOS ONE
---

## [Editor Report · Acceptance letter]

PONE-D-25-03127R1

PLOS ONE

Dear Dr. Briers-Louw,

I'm pleased to inform you that your manuscript has been deemed suitable for publication in PLOS ONE. Congratulations! Your manuscript is now being handed over to our production team.

Kind regards,

on behalf of

Dr. Francesco Bisi

Academic Editor

PLOS ONE